# Axons compensate for biophysical constraints of variable size to uniformize their action potentials

**János Brunner, Antónia Arszovszki, Gergely Tarcsay, János Szabadics** [ID] *

HUN-REN Institute of Experimental Medicine, Budapest, Hungary

* szabadics.janos@koki.hun-ren.hu

**Data Availability Statement:** Raw source data and codes are available in publicly available repositories. Data: https://repo.researchdata.hu/dataset.xhtml?persistentId=hdl:21.15109/ARP/

## Abstract

Active conductances tune the kinetics of axonal action potentials (APs) to support specialized functions of neuron types. However, the temporal characteristics of voltage signals strongly depend on the size of neuronal structures, as capacitive and resistive effects slow down voltage discharges in the membranes of small elements. Axonal action potentials are particularly sensitive to these inherent biophysical effects because of the large diameter variabilities within individual axons, potentially implying bouton size-dependent synaptic effects. However, using direct patch-clamp recordings and voltage imaging in small hippocampal axons in acute slices from rat brains, we demonstrate that AP shapes remain uniform within the same axons, even across an order of magnitude difference in caliber. Our results show that smaller axonal structures have more Kv1 potassium channels that locally re-accelerate AP repolarization and contribute to size-independent APs, while they do not preclude the plasticity of AP shapes. Thus, size-independent axonal APs ensure consistent digital signals for each synapse within axons of same types.

## Introduction

Axons broadcast information about neuronal activity in the form of digital action potentials (APs), whose characteristics align with the specific functions of neuronal cell types. Broad and narrow APs differentially trigger $Ca^{2+}$-influx at synapses, thereby contributing to the cell type-specific strength and plasticity of synaptic responses [1–3]. These precisely tuned presynaptic APs are usually different from somatic APs and they rely on cell type-specific functions of axonal ionic conductances [4–14]. The diversity of axonal $K^+$ channels plays particularly important roles in cell type-specific tuning, determining the speed of repolarization, which controls the duration of synaptic $Ca^{2+}$ channel opening and the driving force for $Ca^{2+}$ influx [4,15,16]. However, in addition to the active components of APs, passive membrane properties also influence AP waveforms. Fast voltage changes are constrained by biophysical principles determined by the small and variable axon diameter [17–20]. Axons are not simple RC circuits, where smaller membrane capacitance compensates for larger input resistance, because the variable surface-to-diameter ratio causes an insistent impedance mismatch. As a result, APs are

BTK8A4 Codes: https://zenodo.org/records/13977231; https://github.com/brunnerjanos/axonal-AP/tree/1.0. All other relevant data are within the manuscript and supporting Information files.

**Funding:** This project has received funding from the European Research Council (ERC) under the European Union's Horizon 2020 research and innovation programme (Grant agreement No. 772452, nanoAXON to J.S.). This work was also supported by the János Bolyai Research Scholarship of the Hungarian Academy of Sciences (J.B. and J.S.). The funders had no role in study design, data collection and analysis, decision to publish, or preparation of the manuscript.

**Competing interests:** The authors have declared that no competing interests exist.

**Abbreviations:** AP, action potential; DTX, dendrotoxin-I; LMFB, large mossy fiber bouton; sMF, small mossy fiber.

expected to be more heavily filtered in smaller axonal structures. The consequential limitations of these filtering effects on the rise and propagation of axonal APs have been recognized for decades, and the accepted view now is that they are compensated by local regulation of $Na^+$ channels and optimal morphology [2,18,21–25] or circumvented by myelin sheaths [26,27]. However, $Na^+$ channels have little contribution to the repolarization phase that is the most critical for the digital output functions.

The duration and decay of APs determine $Ca^{2+}$ influx into the pre-synapse, initiating the translation of digital APs into analog synaptic responses. Due to the fast voltage-dependent operation and efficient activation of synaptic $Ca^{2+}$ channels, AP repolarization-dependent deactivation and driving force control the amount of $Ca^{2+}$ influx, which exponentially determines synaptic release [5,8,28–30]. Hence, even minor changes in AP shape exert a large impact on the evoked responses [2,3,16].

However, the variability in size and geometry of individual synaptic boutons often depends on properties or states that are not directly related to synaptic efficacy, such as the presence of organelles, position along main and terminal transportation routes [31], accumulation of vesicles [21], sudden changes in the membrane surface due to bulk endocytosis [32,33] or excess ionic changes during strong activity [34]. In the synaptic target area, unmyelinated axonal diameters typically range between 0.1 and 5 μm, while synaptic boutons within the same axons also exhibit several-fold differences, resulting in quadratically larger membrane surfaces. Consequently, size variability can substantially complicate synaptic effects by uncontrolled properties that passively influence AP shapes. Therefore, synapse-to-synapse variability in AP decay would not only compromise the cell-type specificity of axonal signaling but also raise questions about whether variable APs within a single axon can truly be considered as real digital signals.

Early theoretical studies predicted that due to the different biophysical properties caused by variable size, the electrotonically isolated axon compartments elicit APs with variable repolarization speed [17,19], implying variable trigger signals for each synapse. However, this question was not experimentally addressed previously. Thus, it remained unknown whether the variability of AP decay between individual boutons of single axons is compensated or provides an additional level of synaptic diversity. Therefore, here we employed direct patch clamp recordings and voltage imaging at unprecedented temporal, morphological, and spatial resolution in various types and sizes of hippocampal axons to understand the size dependence of axonal AP signaling. We focused on the decay phase of APs because it has a fundamental impact on output function of axons by regulating $Ca^{2+}$ influx and has a different correlation to axon size than the extensively studied rising phase, which is crucial for reliable axonal AP propagation [2,18,21–25,35]. Our results provide strong evidence for the maintenance of size-independent AP uniformity within single axons of various types. Among them, in hippocampal mossy fiber axons, Kv1 channel density varies in boutons of different sizes in a manner that compensates for changes in AP shape otherwise expected from capacitive and resistance effects. The varying density of Kv1 channels may be related to distinct anchoring mechanisms and unusual distribution of these channels.

## Results

### Despite variable biophysics similar AP shape is maintained at variable caliber axons

First, we explored the biophysical behavior of variable-caliber axons by simulating APs by sodium and potassium conductances that were homogeneously distributed in the morphology of a real mossy fiber axon originating from a hippocampal granule cell (**Figs 1A and S1**). As expected from simple morphology simulations [17–20], the passive biophysical properties and

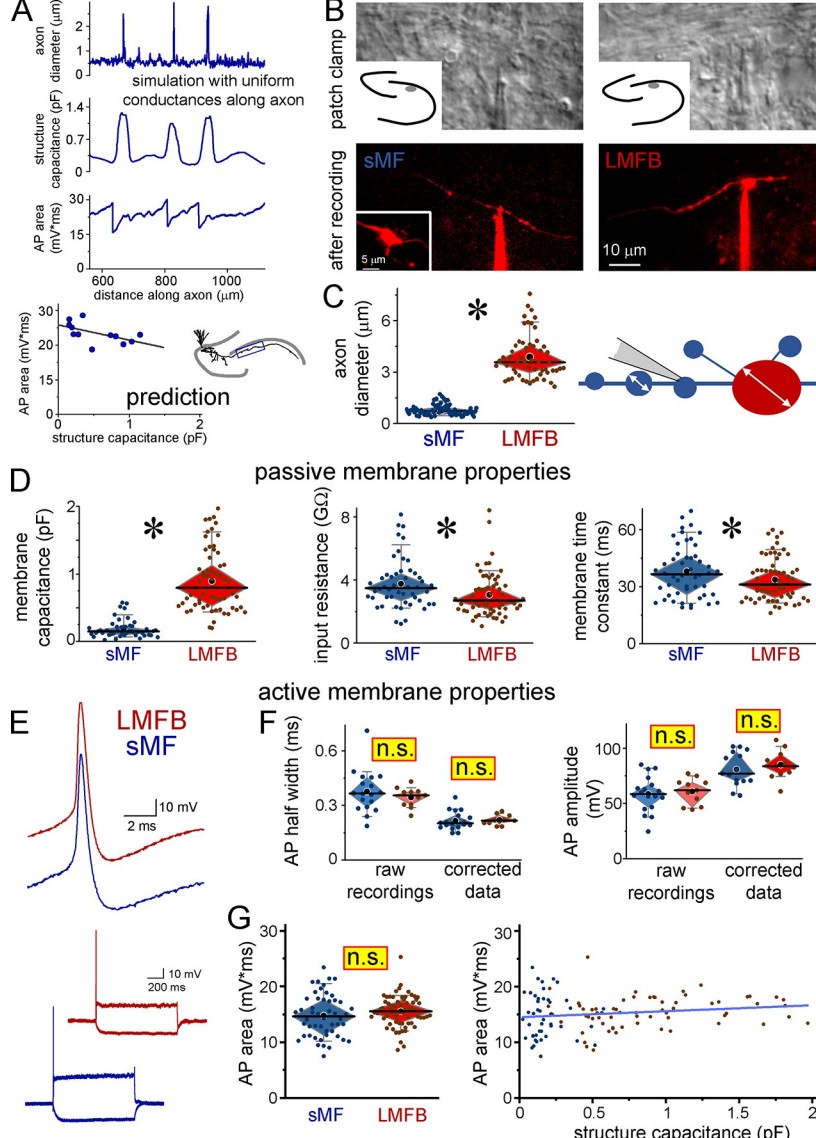

**Fig 1. Similar AP shapes in sMFs and LMFBs despite the different biophysical properties. (A)** Simulations in real MF morphology with uniform $I_K$ and $I_{Na}$ predict that AP kinetics, measured as AP area, is variable due to the size-dependent passive membrane properties. In general, smaller boutons have slower AP. A more detailed and extended versions of these simulations are shown in **S1 Fig. (B)** The recorded sMFs (upper image) were identified with subsequent in situ confocal imaging (lower image) or post hoc anatomy (see Methods). Inset shows an LMFB along the sMF-recorded axon that was used for its identification. LMFBs were recorded in the same conditions as sMFs. **(C)** Their diameters segregated the recorded structures into 2 populations. sMFs include small en passant boutons and main axons. Small dots show individual measurements, large black symbols are the mean, black vertical lines are the median, and diamonds and whiskers show 25%–75% and 10%–90% ranges, respectively. **(D)** The sMFs had smaller membrane capacitance, larger input resistance and slower membrane time constant compared to LMFBs ($n$ = 50 sMF and 64 LMFBs). **(E)** APs were evoked by small current injections. **(F)** AP amplitudes and width at half-maximum (HW) were similar both in the raw recordings and after elimination of the instrumental filtering effect by post hoc simulation of the recording equipment and biological structures [38] ($n$ = 15 sMF and 11 LMFBs). **(G)** The integral area of APs was similar in sMFs and LMFBs and did not correlate with the capacitance of individual boutons, which measures membrane surface and size of the recorded structures ($n$ = 50 sMF and 64 LMFBs). Source data and codes available at https://repo.researchdata.hu/dataset.xhtml?persistentId=hdl:21.15109/ARP/BTK8A4. AP, action potential; LMFB, large mossy fiber bouton; sMF, small mossy fiber.

the waveforms of active conductance-evoked APs changed with the size of the axons. Both the rise and repolarization were generally slower in smaller structures. However, local AP shapes were also influenced by adjacent axonal regions, as sudden widening of the diameter can act as a capacitive sink. While the effects of size and capacitance mismatches on AP propagation (i.e., AP rise/sodium channel activation) have been previously investigated [18,21–26], how these biophysical constraints affect AP repolarization remains unknown. Modeling the subsequent functional steps, the differences in AP shapes were multiplied by the nonlinear correlations between AP shape and $Ca^{2+}$ influx, and between $Ca^{2+}$ influx and synaptic release, resulting in a 40% variability in release efficacy. The results showed that—consistent with previous predictions [17,19,20]—the biophysical consequences of variable axon diameter can lead to changes in AP repolarization in a range that can affect $Ca^{2+}$ influx, which can add complexity to neuronal outputs. Although this simplified simulation did not capture all possible details, it provides a testable prediction: AP repolarization and shape are generally slower in smaller axons, and they are also variable due to mismatch effects at diameter changes.

Next, we tested AP shapes in mossy fibers using direct patch clamp recordings from small axon terminals. These axons form small boutons and connecting segments, whose sizes match most cortical axons, and exceptionally large boutons [3,36]. Mossy fibers lack complex arborization and myelin sheaths, which would complicate their biophysical properties. Only unequivocally identified and intact mossy fibers were analyzed (**Fig 1B**), categorized either as small mossy fibers (sMFs, including both small boutons and main axons) or large mossy fiber boutons (LMFBs) (**Fig 1C**). Cut ends, called blebs [1], were not included. As expected from the smaller size, the membrane capacitance was smaller in sMFs, whereas their input resistance was larger. The measured membrane time constant was also slower in sMFs at resting conditions (**Fig 1D** and **S1 Table**) that would suggest that APs are also size dependent. However, the amplitude and kinetics of locally evoked APs were surprisingly similar in the raw recordings (**Fig 1E and 1F** and **S1 Table**). This may be the consequence of the comparable size of the structures and recording pipettes [1,37–39], which were similarly small in all recordings. As these parameters can be influenced by instrumental filtering caused by the small pipette required for patching axons, we quantified the integral area of APs. Our analysis showed that the integral area of APs was not correlated with bouton identity or the effective size of the recorded mossy fiber structure, measured by local capacitance (**Fig 1G**).

Interference between small axons and experimental instruments affects these recordings [1,37–39]. Therefore, in a subset of experiments, we corrected these signals to investigate the native, instrument-free AP shapes by correcting current clamp errors through composite simulation of the morphology, currents, and the recording instruments [38]. Importantly, the corrected APs of sMFs and LMFBs remained indistinguishable (corrected data in **Fig 1**, **S1 Table**). The smaller variance of the corrected data compared to the raw data confirms minimal biological variability of APs. In contrast to the models that predicted variable AP decay when active conductances were homogeneously distributed, our results using corrected patch clamp data suggested that despite the complex biophysical consequences of variable axon size, APs in sMFs are similar to LMFB APs. Additionally, the corrected voltage data confirmed that AP area measurements are insensitive to instrumental filtering (**Fig 1F**).

## Simultaneous imaging of sMFs and LMFBs confirms AP uniformity within individual axons

To test the AP shapes simultaneously in various axonal compartments, we used fast voltage-sensitive dye imaging which allows simultaneous monitoring of APs in different structures of the same axons at sufficient temporal resolution. Voltage imaging of distal axons was

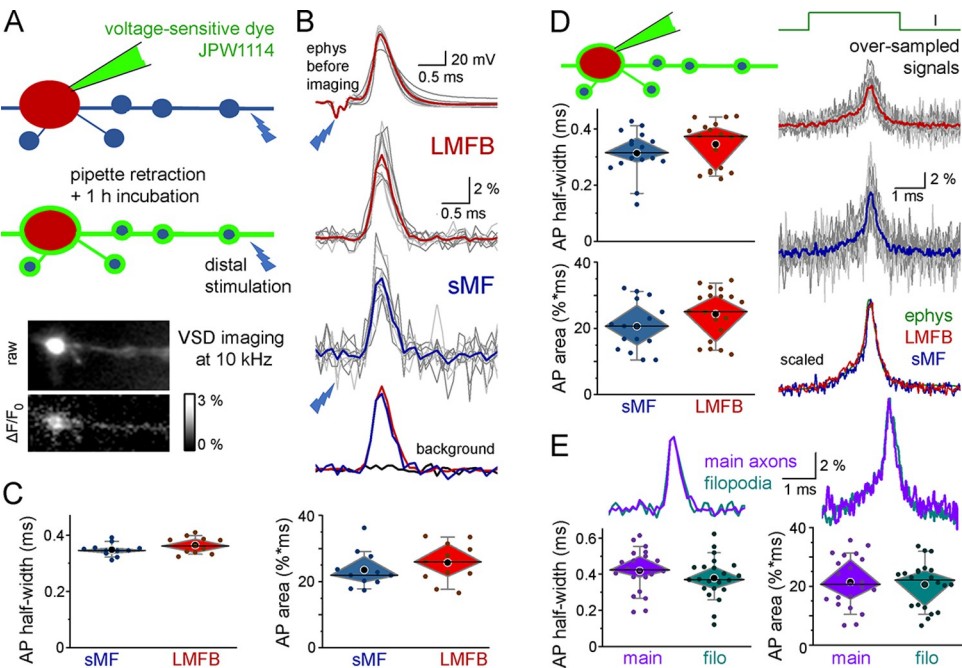

**Fig 2. Voltage-sensitive dye imaging confirms similar AP shapes in sMFs and LMFBs. (A)** JPW1114 (50 µM) was loaded via recording of an LMFB. The pipette was retracted and after equilibration (≥1 h), fluorescence signal changes were monitored at 10 kHz in the vicinity of the recording site, while APs were evoked by distal stimulation (>100 µm). **(B)** Electrophysiological traces were recorded at the beginning of the recordings, independent of the imaging data. Gray traces show individual experiments, with colored average APs from all sMFs and LMFBs ($n$ = 11 experiments). Black trace shows the average signal from pixels right next to the imaged axons. **(C)** AP parameters from simultaneously imaged LMFBs and sMFs. **(D)** In the second imaging approach, the loading pipette was kept on the LMFBs allowing the over-sampling of imaged signals by the high-resolution recordings ($n$ = 18 experiments). Amplitudes are not comparable because the presence of the excess dyes in LMFBs reduce signals by increasing non-responding dye fraction (i.e., $F_0$). **(E)** Voltage imaging revealed no difference between the local APs of 2 distinct sMF structures, the main axon (main) and the filopodial extensions (filo) of the same axons. Source data are available at https://repo.researchdata.hu/dataset.xhtml?persistentId=hdl:21.15109/ARP/BTK8A4. AP, action potential; LMFB, large mossy fiber bouton; sMF, small mossy fiber.

challenging because achieving sufficient concentrations of the membrane bound dye at distances of 100 to 1,000 micrometers results in toxically high concentrations at the somatic loading sites. To overcome these limitations, we directly loaded JPW1114 into LMFBs and their neighboring axon segments with sMFs. This dye is fast enough to resolve the rapid time course of axonal APs [1,11]. First, we investigated naturally propagating APs evoked by distal stimulation. The dye-loading pipette was subsequently retracted (**Fig 2A**). The shapes of the simultaneously imaged APs of sMFs and LMFBs were indistinguishable, and they also matched the previously recorded APs ($n$ = 11; **Fig 2B and 2C** and **S1 Table**).

Because this measurement is limited by the 10 kHz imaging rate, in the next experiments, we used higher temporal resolution with the guide of simultaneous electrophysiological signals (**Fig 2C**). We kept the dye-loading pipette on the LMFBs during imaging and oversampled the JPW1114 signals of locally evoked APs by aligning individual traces to the 100 kHz electrophysiology signals before averaging for distinct axonal compartments. This arrangement reliably reports the temporal profile of APs, although the amplitudes are not comparable due to the presence of the excess, non-responding dye in the pipette on the LMFB (i.e., $F_0$). Importantly, this measurement also confirmed uniform AP shapes in sMFs and LMFBs ($n$ = 18, **S1 Table**). Furthermore, simultaneous imaging revealed similar AP shapes in 2 distinct types of

sMF structures: the filopodia emanating from LMFBs and main axons with small en passant boutons (**Fig 2E** and **S1 Table**). Furthermore, we did not detect differences in the APs of bassoon-positive and -negative small axonal segments (identified by correlated post hoc immunolabeling, **S2 Fig**) suggesting that AP shape is similar in synaptic and non-synaptic segments. Altogether, we concluded from our independent measurements that AP shape is uniform within individual mossy fibers despite the large intrinsic variabilities of their diameters and biophysical properties.

## Cell type-specific but uniform APs in various axons

Next, we tested whether the AP uniformity applies to other cell types. We measured the APs and biophysical properties in diverse types of axons, representing the main cortical axon types, including subcortical and locally originated glutamatergic and GABAergic types. The axons were rigorously identified by using their morphological properties and immunolabeling for 4 to 5 markers on each axon (see **Figs 3A, 3B,** S3, S4 **and** S5). First, we confirmed in these axons that the membrane properties and AP shapes are axon type specific (**Fig 3C and S2 Table**), validating the principle that axonal functions are cell type-specifically tuned. Axons from hypothalamic supramammillary nucleus (SuMa) elicited wider APs than mossy fibers (average AP area: $25.8 \pm 1.2$ mV*ms, $n = 16$), while the axonal APs of hilar mossy cells ($21.6 \pm 3.6$ mV*ms,

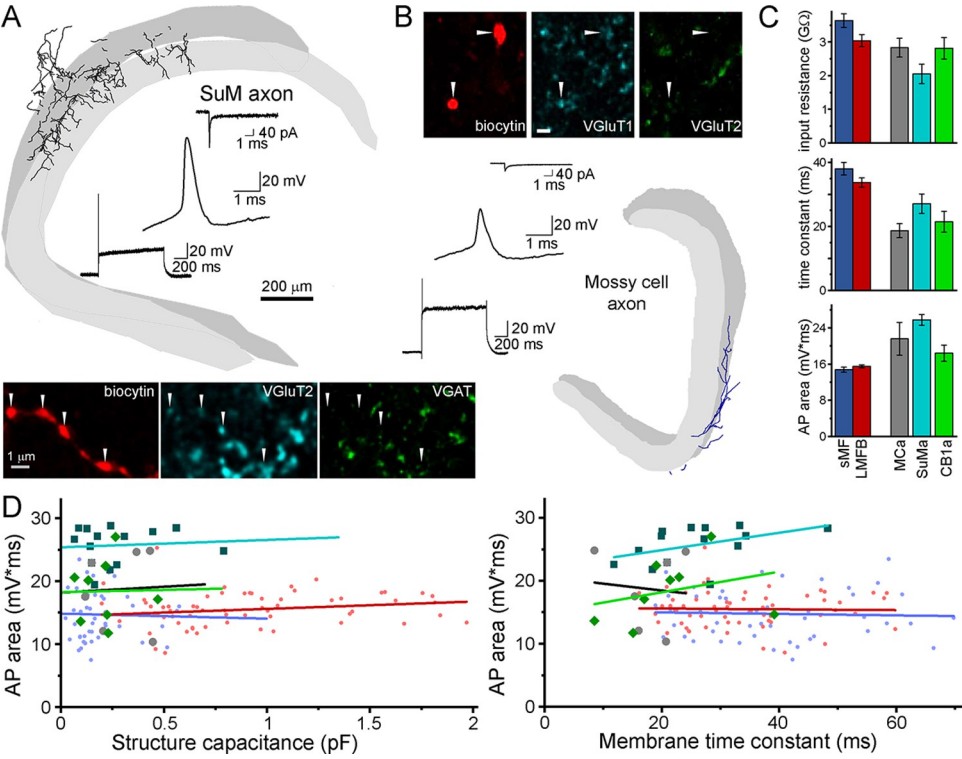

**Fig 3. AP uniformity in diverse types of hippocampal axons. (A)** A representative recording from a morphologically identified VGluT2+ SuM axon in the DG (further details of identification are shown in **S3 Fig**). **(B)** A representative recording from a morphologically identified VGluT1+ mossy cell axon (see also **S4** and **S5 Figs**). **(C)** Average electrical properties of axons are cell type dependent (ANOVA: F(4) = 4.45, 7.5, and 27.4, $p$ = 0.002, $1.6*10^{-5}$ and $1*10^{-16}$, Bonferroni pairwise comparison is shown in **S2 Table**). **(D)** Within cell types, the AP shapes did not show correlation with the local size and membrane time constant of the axons (statistical details of linear fits are shown in **S3 Table**). Source data are available at https://repo.researchdata.hu/dataset.xhtml?persistentId=hdl:21.15109/ARP/BTK8A4. AP, action potential.

$n$ = 7, MCa) and CB1R-expressing GABAergic axons (18.4 ± 1.8 mV*ms, $n$ = 8, CB1a) were between SuMa and mossy fibers (LMFB: 15.5 ± 0.4 mV*ms, $n$ = 64, sMF: 14.8 ± 0.5 mV*ms, $n$ = 50). The cell type-specific APs also demonstrate that the observed uniformity of mossy fiber APs was not due to the insensitivity of recording methods.

These axons, like most in the CNS, have variable diameter due to varicosities. However, within anatomically homogeneous cell types, the AP shape did not depend on the size and passive membrane properties of the recorded axonal structure (**Fig 3D** and **S3 Table**). These results suggest that all tested axon types employ uniform APs in spite of their variable size and biophysical properties.

## Size-independent AP plasticity

Axonal AP waveforms are not always the same within individual boutons and their shapes exhibit functionally relevant plasticity, as axonal APs typically show activity-dependent broadening during sustained activities influencing their synaptic effects [5,6,8,12–14,30,40,41]. To assess whether LMFB and sMF APs maintain uniformity during various dynamic conditions, we conducted direct recordings and VSD imaging (**Fig 4A, 4B, 4D and 4E**). We found that sMF APs, similarly to previous data from LMFBs [5], display frequency-dependent broadening during AP trains. Importantly, the extent of AP plasticity was similar in sMFs and LMFBs and did not depend on the size of the structure (**Fig 4C and 4F**). Thus, AP uniformity persists during dynamic conditions allowing for similar activity-dependent modulation of axonal output.

## Active conductances underlie uniform APs

There must be mechanisms that compensate for the passive biophysical constraints caused by axon size. In the simulations that recreated original recordings of single APs by adjusting the amounts of active currents in the detailed model of an axon with instruments (see **Fig 1F**), both sMFs and LMFBs required a similar amount of sodium currents ($I_{Na}$) on average. However, sMFs needed more potassium currents ($I_K$) than LMFBs. This predicts a lower $I_{Na}/I_K$ ratio in sMFs (**Fig 5A**). To test this prediction, we experimentally measured the $I_{Na}/I_K$ ratio in outside-out patch recordings, where both $I_{Na}$ and $I_K$ were measured in each patch using 2 different voltage steps (**Fig 5B**). Consistent with the model prediction, while $I_{Na}$ densities were similar, the total $I_K$ were larger in patches pulled from sMFs, resulting in larger $I_{Na}/I_K$ ratio ($n$ = 46 sMFs and 47 LMFBs, for statistical details see **S1 Table** and **Fig 5C**). As outside-out patches sample ionic channels from local membranes only, we concluded that sMFs have more $I_K$, which speed up their AP repolarization and promote AP uniformity.

Among axonal Kv channels, Kv1s are known to be responsible for fine-tuning APs for specific functions of various axons [4,6–9,13,14,40,42], whereas Kv3 and BK channels usually provide the backbone of repolarization in mossy fibers [43] and Kv7 channels set axonal excitability that also influence spiking properties [44]. However, the contribution of different Kv channels changes during complex firing patterns. Interestingly, electron microscopic studies noted that Kv1.1 channel distribution was not homogeneous in mossy fibers and the expression was more prominent in small axon compartments [45]. Among these channels, we focused on the potential role of Kv1 channels in uniformizing single axonal APs, we inhibited Kv1.1 and Kv1.2 channels with dendrotoxin-I (DTX), which resulted in wider APs in mossy fibers. However, the effects of DTX were larger on AP repolarization in sMFs than on LMFBs APs (38.6% versus 21.5% at the population level, **Fig 5D**), indicating that sMFs employ more Kv1 channels to achieve fast repolarization. In the presence of DTX, axon size and AP shape showed a similar correlation as predicted in axon models with homogeneous conductances (**S1E Fig**). This observation suggests the non-uniform distribution of AP-relevant potassium

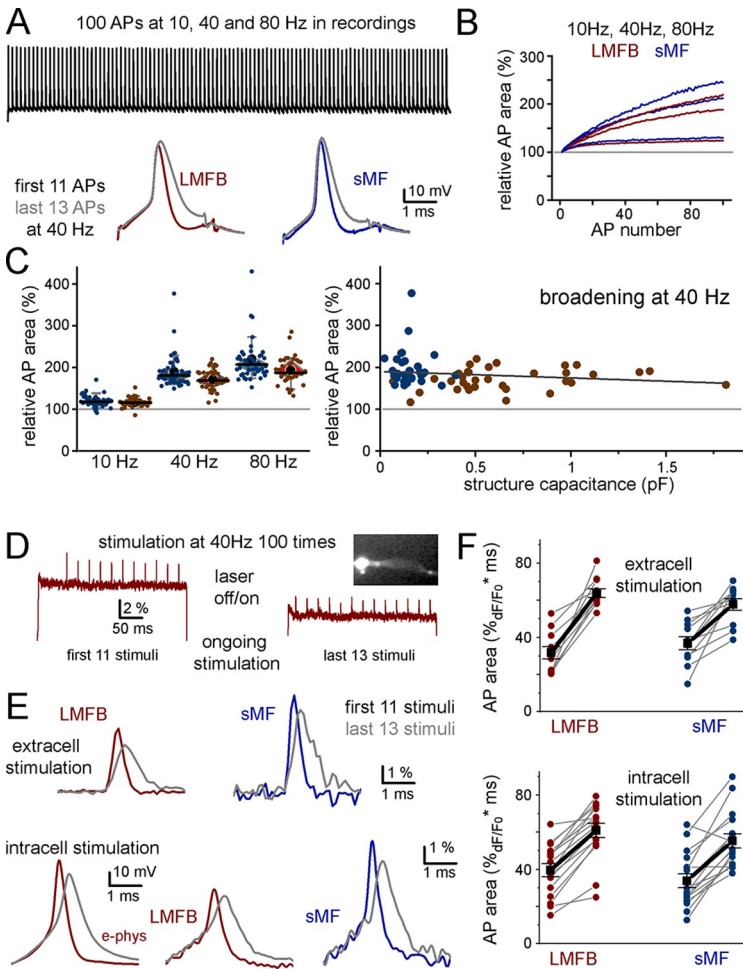

**Fig 4. AP shape plasticity affects sMFs and LMFBs similarly. (A)** AP trains were evoked at 3 different frequencies during direct patch clamp recording of sMFs and LMFBs. The example traces show 40 Hz stimulation, and 3 ms-long current injections pulses were used to avoid the contamination of the repolarization phase by the capacitance artifact. **(B)** Progressive and frequency-dependent broadening of sMF- and LMFB-APs (see also [5]). **(C)** Summary data show similar broadening in sMFs and LMFBs at different frequencies (left panel). Furthermore, the broadening of APs did not depend on the functional size of the boutons (right panel). **(D)** The 100 APs were evoked at 40 Hz in axons during VSD imaging experiments. To avoid photo damage, the axons were illuminated and imaged only during the first 11 and the last 13 APs. **(E)** Example APs obtained by 2 different imaging approaches with extracellular stimulation (upper panels) and with simultaneous intracellular recording, which allowed oversampling of the imaging signals (see **Fig 2**). **(F)** Summary data show similar broadening in sMFs and LMFBs obtained with the 2 approaches. Source data are available at https://repo.researchdata.hu/dataset.xhtml?persistentId=hdl:21.15109/ARP/BTK8A4. AP, action potential; LMFB, large mossy fiber bouton; sMF, small mossy fiber.

currents explains the difference between the measured and predicted size-dependent behavior of axonal APs. It also suggests an important role for Kv1 channels in accelerating APs in smaller MF compartments. Furthermore, the size-dependent DTX effects on APs independently confirmed our patch data about the contribution of larger $I_K$ to the APs of sMFs. Indeed, in the presence of DTX, the $I_K$ density and $I_{Na}/I_K$ ratio were similar in LMFBs and sMFs (**S6 Fig**). The resting membrane potential was not significantly affected by DTX (sMF control versus DTX: −75 ± 0.8 mV versus −72.7 ± 2.6 mV, LMFB control versus DTX: −74.4 ± 1.2 mV versus −75.1 ± 2.5 mV, ANOVA: F(3) = 0.441, $p$ = 0.72, $n$ = 54, 17 and 36, 17), indicating little Kv1 channel activation at resting conditions in MF axons. In contrast to the

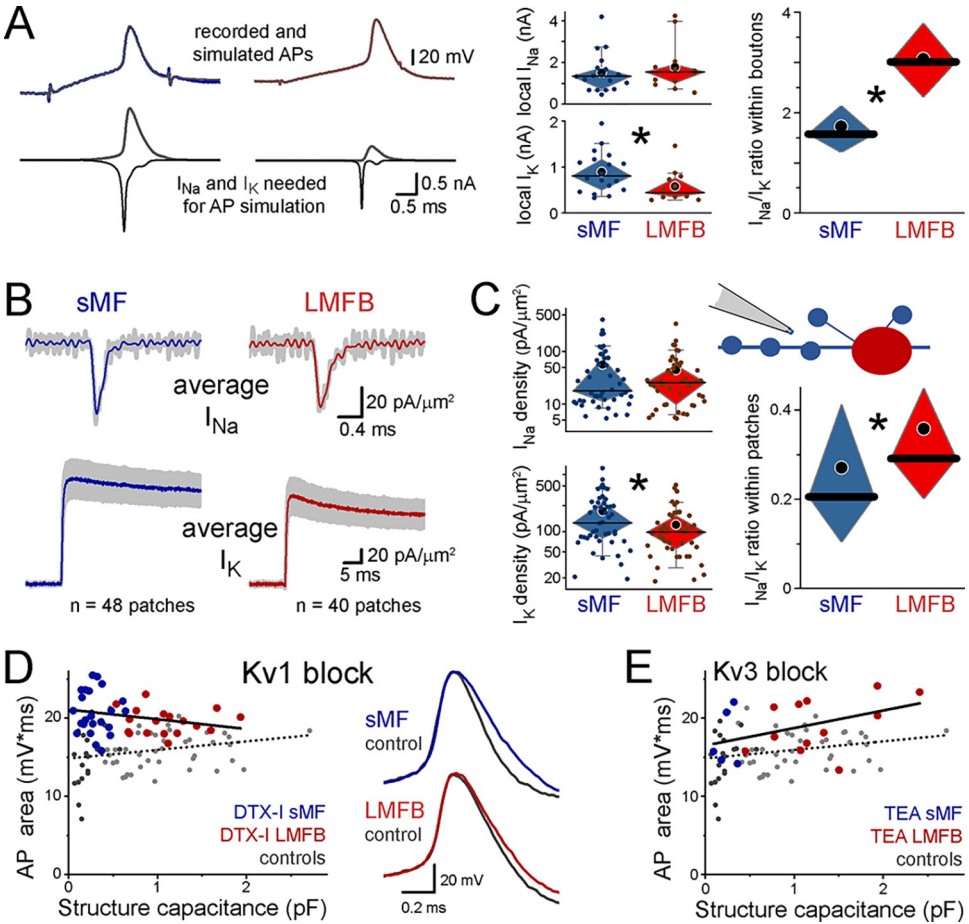

**Fig 5. Active conductances compensate for the biophysical constraints of smaller axon size.** (**A**) Recreating native sMF APs (see **Fig 1**) in simulations required similar $I_{Na}$ but more $I_K$ than in LMFBs, resulting in larger $I_{Na}/I_K$ ratio in the latter. (**B**) Average of recorded $I_{Na}$ and $I_K$ current densities in membrane patches pulled from sMFs and LMFBs ($n$ = 48 and 40 recordings) using 2 voltage-steps in the same recordings (0 mV for 0.4 ms for $I_{Na}$, +70 mV for $I_K$, from −110 mV). (**C**) The average $I_{Na}$ density was similar between LMFB and sMF recordings; however, sMF membranes had larger $I_K$ densities, resulting in larger $I_{Na}/I_K$ ratio in LMFBs. (**D**) Selective inhibition of Kv1-mediated $I_K$ by 100 nM DTX had size-dependent effects on axonal APs with larger effects on smaller sMFs. Four representative recordings are shown on the right. (**E**) In contrast, selective partial Kv3 inhibition by TEA (100 μM) had similar effects on APs of sMFs and LMFBs. Source data are available at https://repo.researchdata.hu/dataset.xhtml?persistentId=hdl:21.15109/ARP/BTK8A4. AP, action potential; DTX, dendrotoxin-I; LMFB, large mossy fiber bouton; sMF, small mossy fiber.

DTX effects, partial inhibition of Kv3 channels by 100 μM of TEA had similar effects on sMF and LMFB APs (16.9% and 19.5%, **Fig 5E**). These results show that the contributions of at least 2 AP-relevant channels are differently related to axon size. While Kv3s have similar contributions to single APs, the contribution of Kv1 correlates with axon size compensating the biophysical consequences of small and variable axon size.

## Discussion

We demonstrated that individual axon types elicit size-independent uniform APs. While theoretical considerations and our initial simulations predicted that AP repolarization would be distorted by the local biophysical environment of varying axon size, our direct axonal patch-clamp recordings and voltage imaging showed that AP repolarization is remarkably resistant to these effects. We also found that Kv1 channels play a crucial role in maintaining axonal

uniform AP shapes regardless of size. These channels accelerate the repolarization phase, compensating for the slower membrane time constant in smaller axonal structures, where their contribution is greater than in larger axons. Having a uniform AP signal in all synapses throughout the axon of an individual cell has several fundamental advantages and consequences. Firstly, with coordinated uniform trigger signals, synapses can individually diversify their output, relying on the known diversity of synaptic machineries that allows pre- and post-synaptic cell alignment-, state-, or learning-dependent communications. If AP shapes varied with the size of the presynaptic boutons, the release machinery would have to adjust for the actual size of each synapse. Homeostatic regulation of the density of a single channel according to the size of boutons and uniform APs can eliminate the need for such complex mechanisms. Notably, Kv1 channels are anchored by molecular pathways that are independent from other AP-mediating channels (see below for details) and these channels have a sufficiently fast activation to accelerate repolarization [7,9,13,43].

Secondly, without uniform axonal APs, the previously described cell type-specific rules of axonal signaling would be compromised. Diverse shapes of axonal APs evolved to fulfill cell type-specific functions. For example, narrow APs cope well with high-frequency firing in the cerebellum [8,12], whereas wider APs mediate reliable release in circuits, where sub-millisecond precision is not required [10,41]. However, the effect size of the biophysical consequences of variable axon diameter on AP kinetics are comparable to differences of APs in various cell types. Thus, if passive consequences of variable size were not compensated, cell type-specific AP shapes could make less significant functional contributions. We showed that AP uniformity applies to diverse cell types serving distinct physiological functions through specific AP shapes. Therefore, it is likely that axonal AP uniformity is a general feature of axons. These results are also important from a technical standpoint, because they confirm that our measurements are sufficiently sensitive for small differences in AP kinetics, and these recordings revealed previously unknown axonal AP properties, including those of hippocampal axons from supramammillary nucleus and hilar mossy cells.

Thirdly, AP uniformity is also important for consistent effects of AP shape plasticity. Activity-dependent AP broadening also falls within the same range as the expected passive effects caused by the variability of bouton size within individual axons. We found that AP uniformity applies not only to solitary APs, but small and large boutons showed similar broadening during AP trains. In this regard, one of the key aspects of AP decay in mossy fibers is the involvement of multiple potassium channels. Repolarization is primarily mediated by Kv3 with additional activity-dependent support from BK channels in LMFBs [43].Kv7 plays a significant role in setting the intrinsic excitability, consequentially impacting AP signaling, especially during complex firing patterns [44]. The exact role of Kv1 channel in mossy fibers was not known. Our findings show that Kv1 channels significantly contribute to AP repolarization, particularly in sMFs. Unlike Kv3, Kv1 functions correlate with bouton size or type during single APs. Kv3 and Kv7 channels share the ankyrin-spectrin pathway for axonal anchoring with sodium channels [46,47]. In contrast, Kv1 channels are anchored through the PSD93-ADAM22-LGI1 pathway and are associated with Caspr2 proteins location dependently [48–50]. Kv1 transport is regulated by unique cargo-tag mechanisms involving CDK5 [51]. The direct interactions of PSD-93 and CDK5 with the cytoskeleton [52] may offer a potential molecular mechanism linking bouton volume to Kv1 density [53,54]. Future studies should also explore the roles of these pathways together with the structure size-dependent stoichiometry and sub-axonal localization of other potassium channels.

Importantly, at least 1 type of potassium channel has size-dependent contributions, while other channels support repolarization in a size-independent manner. Although we focused on Kv1 and Kv3 channels, it is possible that other channels are also regulated by axon size and

their combined contribution is necessary for uniform axonal AP under various physiological conditions. The cell type-specificity of axonal signaling also implies that the mechanisms maintaining uniform axonal APs are likely diverse. In mossy fibers, Kv1 channels play a key role in this homeostatic AP uniformization. This finding is further supported by previous electron microscopic analysis showing that Kv1.1 channels are present in small axons but are generally absent from large terminals [45]. Generally, Kv1 channels are often employed for fine-tuning axonal APs [6,7,12–14,55,56]. Notably, cephalopods overcome slower biochemical mechanisms in low-temperature environments using Kv1 channels with faster opening and closing kinetics by elaborated RNA-editing mechanisms depending on the geographical latitude [57].

We measured APs only from unmyelinated segments of axon. Direct patch clamp recordings are not possible from the myelinated axons. While SuM, MC, CB1R axons are known to have myelinated segments, in the recorded regions they were mostly unmyelinated as also suggested by large number of synaptic boutons. While APs can be measured in myelinated segments with VSD imaging, we employed this method only for mossy fibers, which do not have myelin sheaths. APs are known to be different in the myelin-sheath-insulated and exposed membranes [26], but the insulated segments do not contribute directly to the output functions of axons [2]. Therefore, AP uniformity is not relevant in these segments.

Importantly, several components of Kv1 anchoring and regulatory complexes are directly implicated in diseases. Specifically, several forms of mutations in the Kv1-anchoring LGI1 proteins lead to autosomal dominant temporal lobe epilepsies [48,58,59]. Furthermore, auto-antibodies against this extracellular protein initiate autoimmune encephalitis [60,61], which also inhibit their interaction with the ADAM22-family proteins [59]. Caspr2 proteins are also autoimmune targets, causing periferial neuromyotonia [58]. Episodic ataxias are associated with various mutations in the Kv1.1 gene [62], while loss of Kvβ, which is involved in the normal targeting of Kv1 channels, was observed in patients with severe seizures [58,63]. Axonal swellings are biomarkers of Alzheimer's disease, and they potentially contribute to the development of abnormal neuronal functions in this disease [64,65]. Because previously it was not known that these proteins and features are important for maintaining uniform axonal AP signaling, future studies should clarify the contribution of disrupted size-independent AP uniformity to the development of these diseases.

## Methods

### Ethics statement

All experiments were performed in accordance with the Institutional Ethical Codex, Hungarian Act of Animal Care and Experimentation (1998, XXVIII, section 243/1998) and the European Union guidelines (directive 2010/63/EU), and with the approval of the National Scientific Ethical Committee on Animal Experimentation (NÉBIH, PE/EA/48-2/2020) and Institutional Animal Care and Use Committee.

### Slice electrophysiology

Hippocampal slices were prepared from young rats (P23-35, deeply anesthetized with isoflurane) in ice-cold artificial cerebrospinal fluid (85 mM NaCl, 75 mM sucrose, 2.5 mM KCl, 25 mM glucose, 1.25 mM $NaH_2PO_4$, 4mM $MgCl_2$, 0.5 mM $CaCl_2$, and 24 mM $NaHCO_3$, Leica VT1200 vibratome). We used an upright microscope (Eclipse FN-1; Nikon) equipped with a high numerical aperture objective (25× NA1.1W objective, Nikon) and with a confocal scanning system (C1 Plus, Nikon). Recording solution was composed of 126 mM NaCl, 2.5 mM KCl, 26 mM $NaHCO_3$, 2 mM $CaCl_2$, 2 mM $MgCl_2$, 1.25 mM $NaH_2PO_4$, and 10 mM glucose. Recordings were performed using an intracellular solution containing 90 mM potassium

gluconate, 43.5 mM KCl, 1.8 mM NaCl, 1.7 mM MgCl$_2$, 0.05 mM EGTA, 10 mM HEPES, 2 mM Mg-ATP, 0.4 mM Na$_2$-GTP, 10 mM phosphocreatine, 8 mM biocytin, and 20 μM Alexa Fluor 594. The temperature was kept at 34 to 36˚C during the recordings. The axonal structures in the stratum lucidum of CA3 region or in the inner layer of stratum moleculare of DG region were patched under the guidance of IR-DIC optics by using capillaries pulled from borosilicate tubing (1.5 mm outer, 0.75 mm inner diameter, Sutter Instruments). Data was collected using a MultiClamp700B amplifier and digitized by a Digidata 1440 A/D converter at 250 kHz sampling rate (Molecular Devices). Capacitance neutralization was carefully adjusted close to its possible maximum. After the recording, we captured the position of the recording electrode together with the in situ morphological characteristics of the recorded axon based on the fluorescent Alexa Fluor 594 signal by using the confocal scanning head. For detailed visualization of the morphology, we employed post hoc fluorescent streptavidin reaction. Only identified axons were included (see below) and recordings close to the cut end of the axons were excluded.

For the measurement of the local electrical parameters of axons, 100 ms-long pulses from −70 mV to −90 mV were used in voltage clamp. First, we recorded the signals in cell-attached configuration, which allowed the estimation of the seal resistance in each measurement (12.1 ± 0.6 GΩ for sMF, 13.5 ± 0.6 GΩ for LMFBs, and 12.5 ± 0.9 GΩ for other axons). For the quantification of local axonal capacitance, the average of the cell-attached signal was subtracted from the whole-cell recorded trace to eliminate residual uncompensated instrumental capacitance. We fitted the capacitive current transients with the sum of 2 exponential functions in a 3 to 7 ms time window after the peak assuming that a two-compartmental electrical equivalent circuit is an adequate representation of recorded axons [66,67]. In this model, the faster exponential describes the electrical behavior of the local membrane at the recording site, whereas the slower one is the composite representation of the entire axonal arbor. To validate this two-compartmental simplification, we systematically fitted the traces with mono-, bi-, or tri-exponential functions in a subset of the experiments. The two-compartmental model significantly outperformed the fit of the mono-exponential model assessed (sum of squared error: sMF: 7.81 nA$^2$ versus 2.37 nA$^2$; LMFB: 10.58 nA$^2$ versus 2.18 nA$^2$ for mono- and bi-exponential fits, respectively). Furthermore, fitting the traces with the sum of 3 exponentials resulted in only marginal improvement compared to the biexponential model (sMF: 2.37 nA$^2$ versus 2.16 nA$^2$; LMFB: 2.18 nA$^2$ versus 2.04 nA$^2$ for bi- and tri-exponential fits, respectively) indicating that this simplified two-compartmental model is necessary and sufficient to characterize the passive electrical properties of the recorded axons. Thus, the structure capacitance derived from the fast component of the fit and it represented the biophysically effective size of each recorded bouton. Input resistance was measured both in voltage- and current clamp recordings to check the consistency of the recording, but for quantification we used the voltage clamp data. The membrane time constant data derived from independent current clamp recordings. To calculate the AP area in electrophysiological recording data, we first determined the start of the AP at the voltage value where the upstroke velocity reached 100 mV/ms. Then, we computed the AP area within the region between the start of the AP and the point where the membrane potential returned to the same voltage value.

Membrane patches were pulled from the recorded axons after the recording of whole-bouton properties of the structure, including the structure capacitance. The membrane area was estimated from the correlation between pipette resistance and the capacitance of the outside-out patches. To obtain this correlation, we precisely determined the capacitance of the pulled patches in the subset of experiments by pushing the pipette tip into a Sylgard ball at the end of the recording. This area was used to normalize the recorded I$_{Na}$ and I$_K$ from each bouton. The average estimated areas of outside-out patches were 3.69 ± 0.14 μm$^2$ and 3.84 ± 0.17 μm$^2$ for sMFs and LMFBs.

## Modeling of axonal recordings with realistic pipettes and amplifier circuits (Fig 1F)

Computer simulations were performed in the NEURON simulation environment (v7.5) [68] using a detailed model that captures the experimentally observed behavior of the measuring system as described in Olah and colleagues [38]. Pipette parameters and the compensatory mechanisms of the amplifier were set according to the actual measurements using modified model-cell circuits or short-circuiting components of the amplifier. Furthermore, the pipette parameters were quantified using partial submerging, tip-breaking, and direct microscopic measurements of their physical dimensions. The pipette models were adjusted for each recording pipette to recreate capacitive signals obtained in open and on-cell situations. The amplifier and adjusted pipette models allowed to recreate the characteristic capacitive signals during each current clamp recording.

To incorporate realistic morphologies in the model, we used a 140 μm-long detailed reconstruction of the recorded part for each axon. The reconstruction was performed semi-automatically by the Vaa3D software [69] using the confocal images taken right after the recordings. Remote axonal parts were implemented as simplified dummy compartments. To obtain the passive background of the axon, we fitted model responses to the experimentally recorded sub-threshold voltage signals by optimizing axonal passive parameters and the access resistance of the recording. These fundamental parameters were similar for models which were based on LMFBs and sMFs ($C_m$ = 0.90 ± 0.08 and 0.89 ± 0.05 μF$^*$cm$^{-2}$, $R_a$ = 151.49 ± 18.81 and 134.45 ± 15.66 Ω$^*$cm and $R_{access}$ = 117.81 ± 7.63 and 185.26 ± 34.31 MΩ). $R_m$ was held constant at 50 kΩ$^*$cm$^2$.

To predict the axonal AP waveforms, we incorporated active sodium and potassium conductance models with adjustable voltage dependence and kinetic parameters onto the model and the current clamp output was fitted to the experimental data by freely varying all channel parameters. For each axon, 10 APs from 4 different initializations were fitted and the average of the conductance parameters yielded by these 40 independent optimizations was considered as the best fit options. Fits were performed using the PRAXIS algorithm of the NEURON's built-in Multiple Run Fitter. In addition to recreating the recorded AP signals and instrument related capacitive artifacts, these fits predicted the underlying $g_{Na}$ and $g_K$ properties as well. Finally, predicted native AP signals were obtained by running these constrained $g_{Na}$ and $g_K$ in realistic axonal morphology without the inclusion of the pipette and amplifier circuit models. To generate experimentally testable currents, these $g_{Na}$ and $g_K$ were used to generate currents in response to voltage steps in individual axons.

NEURON codes are available at Zenodo: https://zenodo.org/records/13977231 and Github: https://github.com/brunnerjanos/axonal-AP/tree/1.0.

## VSD imaging

The tip of the pipettes was backfilled with normal intracellular solution (see above) and the voltage sensitive dye (JPW1114 also called di-2-ANEPEQ, 10 to 50 μM [70]) containing intracellular solution was carefully loaded thereafter. LMFBs were patched within 4 min after the loading of the pipette. After forming whole cell configuration, the dye was allowed to diffuse into the axon at least for 30 min before starting the imaging experiment. The upright microscope (Nikon FN-1) was equipped with a 1.1 numerical aperture 25× water-immersion objective and a 1.5× magnifier lens in front of the camera. For imaging voltage signal, we used a single mode 532 nm laser source (CNI Changchun MSL-FN-532S-400mW), which was coupled via a bifurcated SMA-905 fiber guide (0.4 mm, NA 0.39) to the epifluorecent attachment of the microscope, which was equipped a filter cube (ZT532rdc dichroic mirror and 542 nm

long-pass emission filter). When the axonal objects needed to be illuminated for longer time, for example during focusing and searching, we used a diode laser illuminator (89North LDI-7) via the second branch of the fiber guide. The excitation intensity was regulated by the direct control of the output power and/or by the use of ND filters. Optical signals were acquired with a CMOS camera (Redshirt Imaging, Da Vinci-1K) with a frame rate of 10 kHz ($64 \times 46$ pixels, pixel size: 0.79 μm) and digitized simultaneously with the electrophysiological signals by the AD board of the imaging system. At the end of the experiment, the VSD fluorescence was used also for morphological identification.

## Analysis of VSD signals

Data were discarded if the half-width of the APs showed more than 20% increase of its initial value. A custom Fiji macro was used to correct for photo-bleaching in individual pixels of the image stacks using independent double exponential fits. The pixels were clustered into 8 bins for the fitting. To select the active axonal compartments within bleach corrected image stacks, we generated a $\Delta F/F_0$ image, where the intensities around ($\pm 0.2$ ms) the AP peaks were subtracted and divided with the baseline values of each pixel on the image. The identification was helped by the multifocal morphological images that were recorded after the imaging session. In the first imaging approach in which the naturally propagating APs were imaged, the 10 kHz imaging traces were segmented and aligned with custom Python scripts. For the second imaging approach that intended to increase the temporal resolution of the imaging signals, we implemented a shift and mean algorithm (written in Python) to increase the temporal resolution of the recorded signals (adopted from [71]). In brief, the higher sampling rate of the electrophysiology signals registered simultaneously on the DigiData AD-board allowed us to align the spikes with 250 kHz temporal resolution. The oversampled imaging signals were pooled, aligned according to these more precise AP peaks and then resampled by averaging multiple (typically 8) data points resulting in 24.5 to 27.5 kHz final resolution. For demonstration purposes, signals were low-pass filtered using a bidirectional digital Chebyshev-II lowpass filter (f_stop = 8 kHz, Microcal OriginPro). If the imaging experiment included longer axons, VSD signals were averaged in short axonal segments (<5 μm). The signals from these segments were averaged for each axon for comparison of sMF and LMFB. VSD signals are represented and were analyzed without filtering, except for the half-width measurements, where the signals were low-pass filtered (Butterworth) in 2 steps: 10 kHz and 100 kHz for over-sampled signals, 2 kHz and 20 kHz for extracellular stimulation-evoked APs. Area for background measurements (Fig 2B) was selected 3 to 7 pixels distance from the imaged LMFB, which had the brightest emission. Because of the lack of AP signals and intensity changes in the background area, we concluded that neighboring structures (i.e., LMFB and sMF) can be independently analyzed, which were selected with similar distances. AP area was determined as the integral area in a 0.6 ms window starting from 0.05 or 0.1 ms from the peak of peak-normalized AP signals. Normalization was needed because VSD signal amplitude depends on the background intensity, which is influenced by other factors in some cases (e.g., pipette dye load).

## Anatomical identification of whole-bouton recorded axons

Recorded axons were loaded via the pipette with biocytin and in most cases with Alexa Fluor 594. The fluorescent dye allowed in situ imaging of the axons after their recordings and identification of the recordings in subsequent morphological analysis (Nikon C1 scanning confocal head or RedShirtImaging CMOS Da Vinci-1K camera with eplifluorescent Sutter LED illumination). In case of sMF recordings, axons were identified by unbranching axon that run parallel in the stratum lucidum where they form typical LMFBs. Biocytin labeling and subsequent

fluorescent streptavidin reactions allowed a more detailed visualization of the morphology and immunolabeling. Slices were fixed in 2% paraformaldehyde and picric acid solution for at least 20 h and then they were resectioned (40 to 60 μm). Recorded axons were stained for biocytin using Alexa-conjugated streptavidin (1:500, Thermo Fisher, S11227). The recorded axons were typically present in 3 to 5 sections. Short axons were excluded from the analysis. To maximize the capacity for immunolabeling for the recorded axons, each section was individually stained with different combinations of 2 or 3 primary antibodies for 72 h: anti-VGluT2 (guinea pig, 1:2,000, Millipore, AB2251, RRID: AB_1587626), anti-VGluT1 (guinea pig, 1:1,000, Millipore, AB5905, RRID: AB_2301751), anti-VGAT (rabbit, 1:1,000, Synaptic System, 131–003, RRID: AB_887869), anti-CB1R (rabbit, 1:1,000, Cayman, 10006590, RRID: AB_10098690), anti-parvalbumin (goat, 1:2,000, Swant, PVG-213, RRID: AB_2650496), anti-parvalbumin (guinea pig, 1:2,000, Swant, GP72, RRID: AB_2665495), anti-calretinin (goat, 1:2,000, Swant, CG1, RRID: AB_10000342), anti-bassoon (mouse, 1:3,000, Abcam, ab82958, RRID: AB_1860018). Fluorescent dye-conjugated (AlexaFluor488, AlexaFluor647, DyLight405) secondary Donkey antibodies were incubated overnight. Sections were mounted in Vectashield or Prolong Glass. The slices were examined with confocal microscopy (Nikon C2 scanning confocal). Image stacks were acquired initially at 0.08 μm/px (XY) and 0.125 μm (Z direction) resolution (objective 60×, 1.4 NA) and then deconvolved (Huygens software). The resolution was lower for axonal reconstructions (0.12 μm/px XY and 0.3 μm/px Z), which were performed with Neurolucida software (version 2020.2.2, MBF Bioscience LLC, Williston, Vermont, USA, RRID: SCR_001775). Bassoon spots were manually assigned as synaptic sites in individual pixels of VSD-imaged mossy fibers using 3D analysis in Nikon NIS Elements software (version AR 4.13.04).

## Modeling of axonal APs, Ca²⁺ influx, and release rate (Fig 1A and S1 Fig)

To model the relationship between axonal morphology and the AP properties and the evoked synaptic release, a propagating AP was simulated in a complete granule cell reconstruction that had intact mossy fiber trajectory in the CA3 field. This morphology was used previously [38], but here we implemented a more complete electrical simulation on this structure. Both the passive parameters ($C_m$ = 1 μF*cm$^{-2}$, $R_a$ = 150 Ω*cm, and 50 kΩ*cm$^2$) and active conductances (290 mS*cm$^{-2}$ and 17 mS*cm$^{-2}$ for the Hodgkin–Huxley type sodium and potassium channels, respectively) were distributed homogeneously in the entire cell. Equilibrium potentials for the leak, $I_K$ and $I_{Na}$ were set to −80 mV, −90 mV, and 70 mV, respectively; whereas, voltage dependence of the Ca²⁺ current followed the Goldman–Hodgkin–Katz [72]. Ca²⁺ influx was modeled as a compound current produced by the mixture of P/Q, N, and R-type channels with a ratio of 6.5:2.5:1, respectively, as described in LMFBs [73]. Ca²⁺ channel kinetics were adopted with a temperature coefficient of $Q_{10}$ = 2.5 to account for the higher temperature used in this model (35˚C). Release was calculated from the Ca²⁺ influx assuming a conservative 2.5-power relationship. A single AP was triggered through somatic current injection and parameters related to the propagating AP and the evoked Ca²⁺ influx was quantified at every micrometer along the axon. As there is no explicit voltage threshold for a propagating AP, the duration of the AP was quantified as the width at −20 mV. This simulation was performed with a temporal resolution of 250 kHz.

## Supporting information

**S1 Fig. Simple simulations predict large variability of AP shape, axonal Ca²⁺ influx, and synaptic release along the variable caliber axons if homogeneous conductances are present.**
(A) Theoretical rules—derived from passive RC properties of neuronal membranes—defining

the dependence of the speed of neuronal signals, presumably including axonal AP shape, on the size of the structure. (B) APs were simulated in a 3D reconstructed mossy fiber by using homogeneously distributed Hodgkin–Huxley sodium and potassium conductances and shown along the axon with variable diameter. Only the data from the middle section of the axon is shown. $Ca^{2+}$ influx was simulated by using a realistic set of $Ca^{2+}$ conductance [73]. Synaptic release related to the integrated $Ca^{2+}$ influx on the 2.5× power. (C) Variable AP kinetics measured as rate of rise and decay depending on the biophysical size of simulated axon compartment. The rise and decay phases of APs show the expected correlation at population level; however, this correlation is variable between individual boutons due to the influence of neighboring axonal regions. (D) Area of simulated APs along the entire axon demonstrate the large variability and axon size-dependence of AP shape. Simulated APs were measured at every 1 μm and were plotted against the membrane capacitance of the same spots. AP area was the integral of the repolarization phase only to avoid contamination of the measurement by effects on the rise phase of the APs [17–20]. (E) AP area and simulated $Ca^{2+}$ influx at 12 boutons that potentially form synaptic contacts within a short segment (linear fits AP area: $R^2$: 0.249, ANOVA $p >$ F: 0.057, slope: −4.37. $I_{Ca}$: $R^2$: 0.287 ANOVA $p >$ F 0.042, slope: 0.261). Colors of traces scales with the capacitance of the boutons. (F) To demonstrate the reliability of AP area measurements compared to classical AP kinetic parameters, we compared raw electrophysiological recordings of APs with their corrected waveforms (see Fig 1F). The correlation between the half-width of APs (HW) and AP area was weaker in raw AP measurements compared to the corrected APs ($R^2$: 0.389 vs. 0.814, left panels). Furthermore, in theory, HW and AP area values should linearly correlate. However, this correlation was significant only in the case of corrected APs ($R^2$: 0.47), but not in the raw data ($R^2$: 0.017). Raw data and simulations are available at https://repo.researchdata.hu/dataset.xhtml?persistentId=hdl:21.15109/ARP/BTK8A4.
(TIF)

**S2 Fig. AP shapes in axon segments that form synapses and connecting parts.** (A) Synaptic regions were identified by presynaptic marker, bassoon (BSN) positivity. Individual pixels were assigned by matching the post hoc reconstruction of biocytin labeled segments that were subject of VSD imaging. Only neighboring segments were included to avoid the contamination by conduction delay. LMFBs were excluded. (B) Confocal image analysis of the bassoon puncta in the biocytin labeled axon. (C) Average APs from 4 experiments with correlated synapse identification. (D) Comparison of the half-widths of APs from sMF segments identified as synaptic or non-synaptic based on BSN staining. Source data are available at https://repo.researchdata.hu/dataset.xhtml?persistentId=hdl:21.15109/ARP/BTK8A4.
(TIF)

**S3 Fig. Identification of axons patched in the DG originating from the supramammillary nucleus (SuMa).** Two examples are shown with the reconstructions of the biocytin-labeled axons and immunolabeling for typical SuM markers and proteins that are present in other axons. SuM axons restricted to the granule cell layer and inner moleculare layer of the DG (3D-represented by light and dark gray areas). Original images are available at https://repo.researchdata.hu/dataset.xhtml?persistentId=hdl:21.15109/ARP/BTK8A4.
(TIF)

**S4 Fig. Identification of axons patched in the DG originating from the hilar mossy cells (MCa).** Two examples are shown with the reconstructions of the biocytin-labeled axons and immunolabeling for typical MC markers and proteins that are present in other axons. MC axons restricted to the inner molecular layer of the DG. Original images are available at

**S5 Fig. Identification of axons originating from the GABAergic CB1-receptor expressing cells (CB1a).** Two examples are shown with the reconstructions of the biocytin-labeled axons and immunolabeling for typical GABAergic cell markers and proteins that are present in other axons. Original images are available at https://repo.researchdata.hu/dataset.xhtml?persistentId=hdl:21.15109/ARP/BTK8A4.
(TIF)

**S6 Fig. When Kv1 components are eliminated, sMFs and LMFBs have similar $I_K$ currents and $I_{Na}/I_K$ ratio.** (A) Average of $I_K$ current densities in membrane patches pulled from sMFs and LMFBs in the presence of DTX (gray traces) and in control conditions. (B) The average $I_{Na}$ and $I_K$ densities and $I_{Na}/I_K$ ratio were similar in LMFB and sMF recordings in the presence of DTX. Source data are available at https://repo.researchdata.hu/dataset.xhtml?persistentId=hdl:21.15109/ARP/BTK8A4.
(TIF)

**S1 Table. Statistical results of comparison of passive and AP properties of sMFs and LMFBs.** Mean and median data with SEM and Q1–Q3 ranges are shown in figures together with individual data points.
(PDF)

**S2 Table. Statistical results of entire population (ANOVA) and pairwise (Bonferroni) comparison of AP area data of different axon types.** Mean data are shown in **Fig 3C**.
(PDF)

**S3 Table. Statistical data on the correlation between size and membrane time constant with AP shapes.** Linear fits are shown in **Fig 3D**.
(PDF)

## Acknowledgments

We thank Balázs Hangya, Árpád Mike, and Ivan Soltesz for their valuable comments on this manuscript. We also thank Andrea Szabó for the excellent technical assistance and Dr. László Barna and Dr. Pál Vági and the Nikon Microscopy Center at IEM for kindly providing microscopy support.

## Author Contributions

**Conceptualization:** János Brunner, János Szabadics.

**Formal analysis:** János Brunner, Antónia Arszovszki, Gergely Tarcsay, János Szabadics.

**Investigation:** János Brunner, Antónia Arszovszki, Gergely Tarcsay.

**Supervision:** János Szabadics.

**Writing – original draft:** János Szabadics.

**Writing – review & editing:** János Brunner, Antónia Arszovszki, Gergely Tarcsay.

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
