## [Editor Report · Decision Letter 0]

30 Apr 2024

Dear Dr Szabadics, 

Thank you for submitting your manuscript entitled "Axons actively compensate for biophysical constraints of variable size to uniformize their action potentials" for consideration as a Research Article by PLOS Biology.

Your manuscript has now been evaluated by the PLOS Biology editorial staff as well as by an academic editor with relevant expertise and I am writing to let you know that we would like to send your submission out for external peer review.

Once your full submission is complete, your paper will undergo a series of checks in preparation for peer review. After your manuscript has passed the checks it will be sent out for review. To provide the metadata for your submission, please Login to Editorial Manager (https://www.editorialmanager.com/pbiology) within two working days, i.e. by May 02 2024 11:59PM.

Kind regards,

Christian

Christian Schnell, PhD

Senior Editor

PLOS Biology

cschnell@plos.org

---

## [Decision Letter · Decision Letter 1]

14 Jun 2024

Dear Dr Szabadics,

Thank you for your patience while your manuscript "Axons actively compensate for biophysical constraints of variable size to uniformize their action potentials" was peer-reviewed at PLOS Biology. It has now been evaluated by the PLOS Biology editors, an Academic Editor with relevant expertise, and by several independent reviewers. 

In light of the reviews, which you will find at the end of this email, we would like to invite you to revise the work to thoroughly address the reviewers' reports.

As you will see below, the reviewers find your study very interesting and well executed. They raise a number of concerns but these should be addressable without a huge amount of additional experimental data.

Given the extent of revision needed, we cannot make a decision about publication until we have seen the revised manuscript and your response to the reviewers' comments. Your revised manuscript is likely to be sent for further evaluation by all or a subset of the reviewers.

**IMPORTANT - SUBMITTING YOUR REVISION**

*Re-submission Checklist*

*Published Peer Review*

*PLOS Data Policy*

*Blot and Gel Data Policy*

Sincerely,

Christian

Christian Schnell, PhD

Senior Editor

PLOS Biology

cschnell@plos.org

REVIEWS:

Reviewer #1: How do living animals spatially scale properties to maintain a given function across species of different size or, even in the same individuals, in equivalent structures with different size? A straightforward example of this fascinating question is the generation and propagation of electrical signals, in particular the action potential (AP), in neurons. This phenomenon is governed by complex non-linear rules involving activation of ion channels and passive properties that depend on the morphology of the neuronal structures. Given these governing rules, the simple linear scaling of active properties does not compensate for size changes (as the authors confirm in Fig. S1), suggesting either a feedback providing a precise "genetic tuning", or a functional compensation at the level of ion channel scaffolding. However, tackling the problem experimentally is normally far beyond the available possibilities.

The work by Brunner is an original, elegant and ambitious contribution in which this problem was addressed, may be for the first time, in a very rigorous manner. Using the specificity of mossy fiber (MF) axons, including in the same structure small MFs (sMFs) and large MF boutons (LMFBs), the authors unambiguously demonstrate using patch clamp (Fig.1) and voltage imaging (Fig.2) that the AP is preserved in the two sites despite the great difference in morphology, a result that can be only understood in terms of compensation of active conductance. Interestingly, in other axonal structures which are more difficult to examine, no correlation between the AP and the size of the axon was found (Fig.3). The study continues by examining short-term AP plasticity (long-term metaplasticity was not investigated) showing that the change in the AP shape during a train is the same in sMFs and LMFBs (Fig.4). 

The second part of the work focusses on the ion channel compensation necessary to produce the phenomenon. In agreement with the theoretical prediction, the authors find a smaller ratio between sodium and potassium currents in sMF outside-out patches with respect to LMBF outside-out patches (Fig.5A-C and Table S1). Thus, the authors found that inhibition of Kv1 channels by dendrotoxin-I changes the AP shape differently in sMFs and LMFBs (Fig.5D), but this differential effect was not observed when they applied TEA. The authors conclude that the compensatory effect responsible for AP uniformity is exclusive of Kv1 potassium channels. In my opinion this conclusion is not justified and I explain why below.

Before going into details, I want to clarify that this manuscript should be published after some major revision, but additional experiments are not requested. Rather, the authors should critically revise the text and pay attention to the complexity of potassium channels, as well as the possibility that also calcium channels may be involved in the compensation.

MAJOR POINTS

The authors state that MFs express Kv1, Kv3 and BK channels, neglecting Kv7 (Martinello et al. (2015); Neuron 85:346-63 ; Martinello et al. (2019) Commun Biol 2:145). The role of these channels can be directly tested because they can be selectively blocked by XE991 or Linopirdine (Brown and Passmore (2009) Br J Pharmacol 156:1185-95). The authors also claim that Kv1 only are responsible for AP fine tuning whereas BK channels "provide the backbone of repolarization". This statement is not correct. BK channels are co-activated by calcium and depolarization and they principally couple to a calcium source because they have a low affinity for calcium (Berkfeld et al. (2006) Science 314:615-20). In agreement with this, it was demonstrated that BK channels operate close to the peak of APs both in axons (Filipis et al. (2023) J Physiol 601(10):1957-79) and dendrites (Blömer et al. (2024) Front Cell Neurosci 18:1353895). Also, the role of BK channels can be tested using the selective inhibitor Iberiotoxin. Overall, the evidence provided by the author suggests that Kv1 channels contribute to the compensation, but it does not support the claim that this contribution is exclusive.

It is also not convincing that the compensatory mechanisms are only due to potassium channels. If one looks at traces in Fig. 4E, the changes in AP shape include a decrease in the peak and a widening of the duration. The decrease in the peak should be mainly due to a partial lack of recovery from inactivation of sodium channels whereas the widening should be due to a decrease of potassium current during repolarization. In a simple scheme involving only sodium and Kv1 channels during the peak, an equal decrease in sodium current at sMFs and LMFBs without any change in Kv1 would lead to an equal change in the ratio between sodium and potassium currents, resulting in the preservation of the AP shape at the two sites. However, also potassium (and calcium) channels are affected during the train and it seems unlikely that the tuning of Kv1 only at the different sites can maintain the AP shape uniformity. One should also think that axons are subject to long-term plasticity characterized by changes in ion channel densities. In my opinion, it would be more plausible the hypothesis that multiple channels, operating in synergy, compensate for size variability. Multiple channels can assemble in sub-micron structures guided by scaffolding proteins and auxiliary proteins, which would lead to concomitant regulation. The authors should revise the Discussion. The idea that the homeostatic regulation of a single channel would be simpler and more economic for the system might not be true because only one channel would limit the flexibility that is required during the life of the axon. In contrast, the concomitant regulation of multiple channels that are physically and functionally coupled would provide ways to adapt more efficiently to the complexity of physiological signaling. 

Reviewer #2: 

This paper shows convincingly that the action potential in the axons of mossy fibers has

nearly the same shape in parts of the axons with small boutons as in the large boutons,

and that in outside-out patches from the small boutons, there is a larger contribution from

Kv1 potassium channels relative to sodium current. The authors argue that this suggests

a general principle that ion channels are regulated in parts of axons with different sizes to

"uniformize" action potentials.

 From a technical standpoint, this is a remarkable paper. The initial papers from the

Jonas and Geiger labs of recordings from mossy fiber boutons were amazing feats of

electrophysiology, and the recordings here from even smaller structures reach a new level

of experimental achievement. These are very difficult experiments, and they are done with

exceptional attention to detail - for example making a correction for the capacitance pf the

recording pipette - and with exceptional thoroughness. The amount of data and the technical sophistication and completeness of the data reach a level that has rarely been achieved in any electrophysiology paper. The experimental conclusions are very convincing. It is a major contribution to the literature on action potentials. 

 That said, I was not convinced by the conclusion that the authors have discovered a

general principle of action potential uniformity. It is certainly interesting that there is a different combination of channel densities in different regions of the axons, but it does not

seem convincing to me that the reason for this is that action potential shape needs to be kept

exactly uniform. The synapses at the small and large boutons are different in many ways,

e.g. in size of pre-synaptic and post-synaptic region, and small differences in the shape

of the presynaptic action potential would seem to be a very minor factor in determining

the overall pre- to post-synaptic function. I find it hard to believe that if it were possible

somehow to adjust the channel densities so that they were uniform along the axon and

there were then small differences in action potentials that overall circuit function would be

affected very much. For that reason, I was unconvinced by the repeated statement, starting in the title, that the difference in channel densities represents an "active"compensation, as if there must be some sort of feedback mechanism forcing the density of Kv1 channels to be such that action potential width is similar. It seemed to me that this wording ascribes a purposeful agency to axon design that may not be warranted. I would offer that as one reader's reaction and leave it to the authors whether to keep this wording, which to me seemed a bit provocative.

 For example, an alternative interpretation of the results could be simply that the density of Kv1 channels is higher in the non-bouton parts of the axon than in boutons because of factors in bouton formation having nothing particularly to do with fine-tuning AP width. If so, there is unlikely to be an all-or-none transition between "axon" properties and "bouton" properties on the scale of a micron (or even less, since some axons are as small as 0.1 micron), it may just be that most of the membrane in small botouns is less different from normal "axon" properties than that in larger boutons. In fact, the recordings from "SMF"s included recordings from the main axon as well as small boutons. So an alternative interpretation could be that bouton membrane has a lower density of Kv1 channels than axon proper -there are obviously many differences between bouton membrane and axon membrane, including pre-synaptic release machinery, probably uptake transporters, calcium chanel dendity, etc - having nothing to do with a requirement to force the AP width to be uniform between boutons and axons.

 Although the paper is in general very well-written, there was one notable omission in

referring to the literature on variability of action potential shape in cells bodies, axons, and

presynaptic terminals -the Hoppa et al. paper (Neuron. 84:778-89, 2014) showing that

the action potential size is much smaller in small presynaptic boutons than in cell bodies,

as a consequence of increased density of Kv1 and Kv3 channels, and suggesting this is

important to allow increased sensitivity of transmission to changes in AP shape. The idea

that the AP shape is actively constrained at presynaptic terminals seems at odds with the

idea that there must be uniformity of action potential shape. Of course, in both that paper

and this manuscript, the essential information is the experimental observations of AP size

and shape in different cell regions, and the speculation of particular reasons this may be important remains hard to prove. 

 Altogether, the authors are to be congratulated on a technically remarkable tour-de-force description of action potentials in axons and boutons. I was not convinced that the data justifies the conclusion that there is an "active" feedback process that is designed to ensure uniformity of action potentials. If it were my paper, I would soften this conclusion. However, I believe that should be the authors' choice. 

Reviewer #3: In the paper by Brunner and colleagues, the researchers present findings from both electrical and optical measurements made from hippocampal axons. They explore the variability of action potentials (APs) within axons, particularly in relation to the diverse geometrical properties of subcellular domains, and how these APs differ from one axon to another. The authors' successful identification of various axon types is noteworthy, and they have amassed a unique and invaluable dataset. The electrophysiological experiments are well-conducted, and the computational model is meticulously developed, providing ample and multiple lines of evidence to convincingly argue that APs travel uniformly within a single type of axon. However, my primary reservation lies in the ambiguity surrounding the so-called 'theoretical rule'. Both the theoretical foundation and the main conclusion could benefit from a more thorough biophysical explanation, and there is a pressing need to further our understanding of how Kv1 interacts with passive membrane properties.

1. Supplemental figure S1 is in my opinion a key figure to understand the scientific rational and deserves to be a main figure. The authors show that with variable axon diameters but uniform voltage-gated conductances the cable properties underlie local AP variability. The description this result section is muddled (lines 62-74) and the authors only mention there is "variability" but without offering testable predictions or a detailed explanation for the biophysical properties contributing to the AP. It seems to me that the LMFB exhibit both a fast rise and fast decay rate of the membrane potential, illustrated by the linear relationship between rise- and decay rates. In the face of uniform voltage-gated sodium and potassium conductances, how can a small capacitance domain, both slowing the AP up- and downstroke rate? If a 2.5 kV/s upstroke is associated with a -0.5 kV/s decay rate there may be a passive contribution for normalizing the AP area; an increased depolarization rate simply activates more efficiently potassium conductances, speeding repolarization. How is Kv1 slowing the upstroke of the AP? More detailed insights into the role of the electrotonic properties to the AP in a uniform model should be explored and properly explained. For example, the authors could plot conduction velocity or the dV/dt decay rate as a function of somatic distance to better show the rule how bouton size shapes the local AP. The parameter AP area is ambiguous and not helpful to explain the biophysical basis. 

2. The tentative conclusion-like arguments (last sentence p.3) on anchoring of Kv1 channels is wildly speculative. Furthermore, about 50% of the discussion is about anchoring of Kv1 channels (PSD 93, ADAM and Caspr2, p. 8-10) and the authors even adventure into the literature on RNA editing of Kv1. However, data to substantiate the differential distribution of anchoring is completely absent and no experimental examination has been conducted to uncover an axon size-dependent clustering and its impact on action potentials. Rather than conjecture, the authors should provide a robust biophysical explanation of the results and cautiously elucidate how a domain with small capacitance leverages Kv1 to affect the properties.

Minor comments

The accuracy of citing the literature needs to be improved. Just one example, refs. #46 and #47 are not directly comparing large versus small boutons as claimed (lines 185, 249). The publications use EM and qualitatively reveal specific ion channel isoform distribution across compartments. 

"Speed of passive voltage changes was also slower in sMFs" (p. 4). The membrane properties were determined in normal extracellular recording solution and thus not passive. "Resting membrane properties" is the correct terminology. 

Please explain why the integral of the AP is a better parameter. For Kv-related AP properties the authors need to convert the voltage-time data into the first temporal derivative.

"This suggests that AP uniformity is a fundamental feature of axonal APs, beyond they plasticity and passive biophysics" (p. 9). Hard to understand and an awkward sentence. Please rewrite. 

"the amount of Kv1 channels correlate with bouton size" (p. 9). Correlation implies continuous variables are plotted against each other. Where are these data?

---

## [Decision Letter · Decision Letter 2]

17 Oct 2024

Dear János,

Thank you for your patience while we considered your revised manuscript "Axons compensate for biophysical constraints of variable size to uniformize their action potentials" for consideration as a Research Article at PLOS Biology. Your revised study has now been evaluated by the PLOS Biology editors, the Academic Editor and the original reviewers.

In light of the reviews, which you will find at the end of this email, we are pleased to offer you the opportunity to address the remaining points from the reviewers in a revision that we anticipate should not take you very long. We will then assess your revised manuscript and your response to the reviewers' comments with our Academic Editor aiming to avoid further rounds of peer-review, although might need to consult with the reviewers, depending on the nature of the revisions.

In addition to addressing the remaining reviewer comments, we would also like you to address the following editorial requests:

* Please add the links to the funding agencies in the Financial Disclosure statement in the manuscript details.

* DATA POLICY:

Regardless of the method selected, please ensure that you provide the individual numerical values that underlie the summary data displayed in the following figure panels as they are essential for readers to assess your analysis and to reproduce it: 1CDFG, 2CDE, 3C, 4CEF, 5AC, S2D and S6B

* CODE POLICY

* Please note that per journal policy, the model system/species studied should be clearly stated in the abstract of your manuscript. 

**IMPORTANT - SUBMITTING YOUR REVISION**

*Resubmission Checklist*

*Published Peer Review*

*PLOS Data Policy*

*Blot and Gel Data Policy*

Sincerely,

Christian

Christian Schnell, PhD

Senior Editor

PLOS Biology

cschnell@plos.org

REVIEWS:

Reviewer #1: The authors addressed my previous points satisfactorily, in particular by toning down some conclusions that are not supported by the current results. In my view the manuscript can be published in the present form. 

Reviewer #2: The revisions have addressed all the major issues raised by the referees.

Although the manuscript is generally written clearly, there were a few cases where the phrasing seemed unclear. 

54 " Our results provide strong evidence for the maintenance of size-independent AP uniformity within 

single axons of various types. Among them, hippocampal mossy fiber boutons use Kv1 channels for this compensation of the consequences of their variable size, consistently with the distinct anchoring mechanisms and unusual distribution of these channels."

The second sentence might be phrased slightly more clearly as " In hippocampal mossy fiber axons, Kv1 channel density varies in boutons of different sizes in a manner that compensates for changes in AP shape otherwise expected from capacitative and resistance effects. The varying density of Kv1 channels may be related to distinct anchoring mechanisms and unusual distribution of these channels." 

193 " Among axonal Kv channels, Kv1s are known to be responsible for fine-tuning APs for 

specific functions of various axons [4, 6-9, 13, 14, 40, 42], whereas Kv3 and BK channels usually provide the backbone of repolarization in mossy fibers [43] and Kv7 channels set axonal excitability that also influence spiking properties [44]. However, their contribution changes during complex firing patterns"

The second sentence could be interpreted as referring to the contribution of Kv7 channels. I think the meaning might be clearer as "The contribution of different Kv channels changes during complex firing patterns".

Reviewer #3: The manuscript has been greatly improved and offers more insight. While the biophysical basis has been clarified better I have still a few minor concerns. 

About the passive normalization the authors state their "pharmacological results are contrary to this alternative hypothesis since the AP area would be uniform even in the presence of dendrotoxin in this case. We highlight this argument at lines 213-215" I can't find such explanation in the text. The authors should offer academic explanations how to explain the contradiction between theory and experiment. 

It would help showing directly whether the smaller boutons have locally larger membrane time constants in the face of constant channel densities. FigS1A lists claims which are not shown but assumed. Note, some Kv1.x channels have also an impact on the resting membrane properties (e.g. Vm) and so it would be good to report such potentially important roles of Kv1.x channels for the MF axon. 

Line 204 "biophysically limited structure". Unclear what the authors mean with this. 

Line 284. 'Direct patch clamp recordings are not possible from the myelinated parts of SuM, MC, CB1R axons'. What is the evidence those axons are myelinated? The images in Figures S3-S5 show a high densities of release sites and, as the authors rightly stated above in the manuscript, myelin would limit presynaptic bouton formation.

---

## [Editor Report · Decision Letter 3]

6 Nov 2024

Dear János,

Thank you for the submission of your revised Research Article "Axons compensate for biophysical constraints of variable size to uniformize their action potentials" for publication in PLOS Biology. On behalf of my colleagues and the Academic Editor, Eunjoon Kim, I am pleased to say that we can in principle accept your manuscript for publication, provided you address any remaining formatting and reporting issues. These will be detailed in an email you should receive within 2-3 business days from our colleagues in the journal operations team; no action is required from you until then. Please note that we will not be able to formally accept your manuscript and schedule it for publication until you have completed any requested changes.

When you attend to those requests, please also do not forget to add the species to the abstract ("in acute slices from rat brains" for example) and a reference to the source data in all relevant figure legends and not only in the legend of Figure 1.

PRESS

Sincerely, 

Christian

Christian Schnell, PhD

Senior Editor

PLOS Biology

cschnell@plos.org